# Current Methods for Body Fluid Identification Related to Sexual Crime: Focusing on Saliva, Semen, and Vaginal Fluid

**DOI:** 10.3390/diagnostics10090693

**Published:** 2020-09-14

**Authors:** Koichi Sakurada, Ken Watanabe, Tomoko Akutsu

**Affiliations:** 1Department of Forensic Dentistry, Tokyo Medical and Dental University, 1-5-45 Yushima, Bunkyo-ku, Tokyo 113-8510, Japan; 2First Department of Forensic Science, National Research Institute of Police Science, 6-3-1 Kashiwanoha, Kashiwa, Chiba 277-0882, Japan; k-watanabe@nrips.go.jp (K.W.); tomoko@nrips.go.jp (T.A.)

**Keywords:** body fluid identification, immunochromatography, DNA, RNA, sexual

## Abstract

Although, DNA typing plays a decisive role in the identification of persons from blood and body fluid stains in criminal investigations, clarifying the origin of extracted DNA has also been considered an essential task in proving a criminal act. This review introduces the importance of developing precise methods for body fluid identification. Body fluid identification has long relied on enzymatic methods as a presumptive assay and histological or serological methods as a confirmatory assay. However, because the latest DNA typing methods can rapidly obtain results from very small and even old, poorly preserved samples, the development of a novel corresponding body fluid identification method is required. In particular, an immunochromatographic method has been introduced to identify saliva and semen from sexual crimes. In addition, for vaginal fluid identification, attempts have been made in the past decade to introduce a method relying on body fluid-specific mRNA expression levels. At present, the development of molecular biological methods involving microRNA, DNA methylation, and resident bacterial DNA is ongoing. Therefore, in criminal investigations, body fluid identification is an essential task for correctly applying the results of DNA typing, although further research and development are required.

## 1. Introduction

DNA analysis is used in most countries around the world to obtain individual genetic profiles and identify persons from biological samples linked to crimes for use in criminal investigations. At present, the short tandem repeat (STR) approach the main method used, which involves determining the number of repeats of several bases in multiple loci contained in the nuclear DNA of cells. In recent years, along with growth in the number of loci registered in the Combined DNA Index System (CODIS) DNA database [1], reagents with a large number of target loci have become widely used, such as the PowerPlex^®^ fusion (Promega, Madison, WI, USA) and the GlobalFilerTM PCR amplification kit (Life Technologies, Carlsbad, CA, USA). In particular, the GlobalFilerTM reagent amplifies 21 autosomal STR loci, DYS391 locus, Y indel locus, and amelogenin locus. Fujii et al. [2] reported that the probability of identity of these 21 autosomal loci obtained using GlobalFiler was 1.84 × 10^−25^ in Japanese population, which was remarkably higher than that (1.8 × 10^−17^) of the 15 autosomal loci obtained using AmpFLSTR Identifier kit (Life Technologies) [3].

However, it is often difficult to extract nuclear DNA from hair or from degraded samples. An alternative is mitochondrial DNA, which is present in large quantities in cells. The individual identification level of mitochondrial DNA is far below the STR approach [4], but effectively augments the value of other information used for investigations. In addition, DNA methodologies, such as Y-STR typing, single nucleotide polymorphisms, and next-generation sequencing technology have also been applied to various practical tasks [5,6,7]. Due to its reliability, DNA is widely recognized, not only by investigators, but also by the general public, as essential in criminal investigations.

The number of cells constituting the human body had previously been estimated to be about 60 trillion. However, the latest report by Sender et al. [8] of the Weizmann Institute of Science in Israel obtained a total of about 30 trillion, almost half of the previous estimate, and it is considered that the estimated number of cells will one day be further revised downward. In any case, we can safely say that the body is composed of a huge number of cells. As all of these cells have the same DNA, when body fluid is the residual crime-related sample used, the DNA origin is essential in proving the criminal act. In particular, in sexual crimes, the identification of saliva, semen, and vaginal fluid is often required, and biochemical, histological, or instrumental analysis methods that focus on the characteristic proteins and cells in each type of body fluid have long been used. However, detection sensitivity and specificity are greatly affected by the type of body fluid, methodological differences, and differences in the environment in which the sample is left. To keep pace with recent improvements in the discriminating power of DNA typing, precise body fluid identification is required now more than ever.

In this paper, we review the current methods for identifying body fluids related to sexual crime, and we discuss both conventional methods and the latest serological and molecular biological methods, focusing on saliva, semen, and vaginal fluid.

## 2. Flow of Forensic Examinations Using Biological Samples

The basic workflow for forensic examinations using biological samples involves the following: Visual examination, presumptive assay, confirmatory assay, blood typing, and DNA typing. The procedure and the examination name vary slightly depending on the type of body fluid to be examined and the condition of the sample. We will briefly describe the workflow for bloodstains as they are the most frequently examined samples. First, the visual condition, which includes color tone, size, and age, is examined. This is followed by the bloodstain presumptive assay, which is a chemical method that exploits the peroxidase-like activity of heme in hemoglobin. The leucomalachite green assay [9], the luminol assay [10], and the tetramethylbenzidine assay [11] are also widely used. The BLUESTAR^®^ FORENSIC kit (BLUESTAR^®^ Forensic, Monte Carlo, Monaco), which contains the luminol assay, is also available. The luminescence of this kit lasts longer than that of the normal luminol method and it does not require complete darkness.

Next, human blood identification is performed as a confirmatory assay. Previously, a serological assay using anti-human hemoglobin precipitin was used, but at present, the main method is a simple and rapid immunochromatography assay using anti-human HbA_0_ mouse antibody. BLUESTAR^®^ OBTI (BLUESTAR^®^ Forensic) is commercially sold and widely used as a forensic kit. However, OC-Hemocatch S (Eiken, Tokyo, Japan), a clinical reagent that is commercially available as a kit for detecting human HbA_0_ in fecal matter, has been introduced in Japan. The forensic utility of this reagent has been fully verified [12], and its cost performance is also favorable. Accordingly, it is actively used in criminal investigations. Clarification of whether the sample concerns menstrual blood [13,14,15,16,17], maternal blood [18,19,20], fetal blood [21,22], or postmortem blood [15,23] is also often required, and this identification is performed through assays that target proteins or expressed genes that are characteristic of each blood type.

After both the presumptive assay and the confirmatory assay show positive results, ABO blood grouping and finally DNA typing are performed. Various blood grouping methods, including that of the ABO blood type, were used for person identification when DNA typing was not available. However, ABO blood grouping is now often omitted, and it sometimes provides valuable information for narrowing down the subjects for DNA typing because the ABO blood group can be determined, not only from bloodstains, but also from saliva, semen, and hair [24,25,26]. As mentioned above, DNA typing is based on the STR type and sex determination on the amelogenin gene, and there is no doubt about their personal identification power.

## 3. Saliva Identification

For saliva identification, the amylase assay, which combines a presumptive and confirmatory assay, is the most important test. Although, α-amylase is present in all body fluids, the α-amylase activity of saliva is much higher than that of other body fluids [27,28] and it has, thus, been regarded as essential for saliva identification. Many animals also have α-amylase activity [27,29]: Mice, rats, guinea pigs, and squirrels have relatively high activity, whereas dogs, cats, cows, goats, sheep, and horses have little to no activity. Currently, a method using a reagent called blue starch, which is a blue dye chemically bound to starch, is used for the amylase assay. Many countries use the Phadebas method [30], based on the Phadebas^®^ Amylase Test (Magle Life Sciences, Cambridge, MA, USA). This method uses a spectrophotometer to measure and quantify the blue pigment dissolved from blue starch by the α-amylase in saliva.

In contrast, a unique method called the blue starch agarose plate method [31], has been used for many years by investigators in Japan. In this method, powdered blue starch is mixed with a hot agarose solution to form a flat gel. A saliva stain sample cut to a few millimeters or less is directly placed on the flat gel, and the dissolution of the blue pigment is visually observed. The important point is that the assessment needs to be made within a reaction time of approximately 30 min to 1 h at 37 °C, which is the optimum temperature of the enzyme, and within a maximum of 2 h when the amount of saliva attached is extremely low [32]. In addition, the reagent SALIgAE^®^ (Abacus Diagnostics, West Hills, CA, USA), which turns yellow when the substrate is decomposed by α-amylase in saliva, is also effective in saliva identification [33].

Furthermore, identification with a more specific method may be required. The serological methods that have been so far used for this purpose, include immunodiffusion and immunoelectrophoresis using polyclonal antibodies, such as anti-human salivary precipitin, anti-human salivary α-amylase precipitin, or anti-human sIgA precipitin [34]. However, it has become difficult to obtain high-titer polyclonal antibodies in recent years, and the frequency of their use is now extremely low. We have developed an enzyme-linked immunosorbent assay (ELISA) targeting statherin as a new marker in saliva and attempted to introduce it into practice [28]. Although, statherin is slightly less sensitive than α-amylase in ELISA, it is highly specific and effective for saliva identification. Notably, statherin is found at high concentrations in nasal secretions [35]. Another saliva confirmatory test is the RSID™-Saliva (Independent Forensics, Lombard, IL, USA) kit for forensic science, which involves immunochromatography for human salivary α-amylase. It is widely used because it can be tested quickly and easily [36].

Research on saliva identification using genes has also been actively conducted since the beginning of the 2000s. In particular, real-time RT-PCR using the mRNA expression levels of salivary-specific proteins has been reported [37,38,39]. We have also confirmed the usefulness of the *STATH* and *HTN3* genes [40,41] and performed various studies to introduce the method into practice [42,43,44]. In addition, methods focusing on microRNA [45,46,47,48,49,50] and DNA methylation [51,52,53,54,55,56] have been actively explored for saliva identification. At present, it would seem that further studies are necessary before microRNA and DNA methylation methods can be applied to practical work. Meanwhile, studies focusing on bacterial DNA such as *Streptococcus salivarius*, which parasitizes in the oral cavity, have also increased [57,58,59,60]. Such methods include the development of a simple and rapid method involving the loop-mediated isothermal amplification (LAMP) technique, as well as a useful method for deteriorated salivary stain samples.

## 4. Semen Identification

Semen is the most frequently examined body fluid in sexual crimes. Semen gives off a unique odor when it is fresh, whereas dried semen stains are pale yellow to gray in color and sometimes stiff and lustrous. However, when the amount and site of semen deposition is unclear, it is helpful to use a variable wavelength light source device such as a Polilight (Rofin Australia, Victoria, Australia).

Semen consists of sperm and seminal plasma. First, a presumptive assay is performed to detect acid phosphatase [61], which is found at high concentrations in seminal plasma. The acid phosphatase test has long been performed via colorimetric tests, such as a method using α-naphthyl phosphate and Fast Blue B and a method using phenolphthalein diphosphate and ammonia [62]. When the test is positive, the former develops a purple color that is stable for a long time, whereas the latter develops a reddish pink color that fades in a relatively short time. SM test reagent (Wako, Osaka, Japan), a kit that applies the former method, is commercially available and widely used in sexual crime investigations.

If the presumptive assay is positive, a sperm test is performed as the confirmatory assay. Microscopic examination involving tissue staining is generally performed. Various staining methods are used, such as Baecchi, Corin-Stockis, Christmas tree (Oppitz), and hematoxylin and eosin staining [63,64,65,66]. In particular, in Christmas tree staining, sperm show a red head and a green tail that are readily visualized [62]. A useful approach when the sample is old or difficult to distinguish from bacteria and dust is SPERM HY-LITER™ (Independent Forensics, Lombard, IL, USA), which is a commercially available kit involving a fluorescent label [67].

At the same time, serological methods using antibodies, can be applied to provide proof of seminal plasma. In particular, in the case of azoospermia or oligospermia, proof of seminal plasma is essential. Immunodiffusion and immunoelectrophoresis using high-titer anti-human semen precipitin [34] have long been used, but now an immunochromatography kit is the technique of choice. Prostate-specific antigen (PSA) is the oldest marker of seminal plasma [68], and SERATEC^®^ PSA Semiquant (Seratec, Gottingen, Germany), which reacts to 1 million-fold diluted semen, has been frequently used as a highly sensitive kit [69]. However, PSA is also expressed in women’s periurethral glands [70,71,72], especially when they are on contraceptives, and it is highly possible to be tested PSA-positive without intercourse [73], indicating that caution is required when PSA is being evaluated alone without sperm. In contrast, seminogelin, derived from seminal vesicles, is a semen-specific antigen that is not detected in females [74], making it a reliable option for determining the presence of seminal plasma. RSIDTM-SEMEN (Independent Forensics, Lombard, IL, USA) is also commercially available as a kit containing anti-seminogelin antibody [75].

Many studies of semen identification using real-time RT-PCR have been reported [37,38,39]. In particular, SEMG1 and PRM2 are effective target genes [40] with high specificity. As PRM2 is expressed in the sperm itself, it will not be expressed in the semen of azoospermic individuals. In addition, as for saliva, there are many reports on the use of microRNA [45,46,48,49,50] and DNA methylation [51,52,53,54,55,56,76,77] in semen identification.

## 5. Vaginal Fluid Identification

Unlike other body fluids, vaginal fluid does not have characteristic proteins for identification, making it difficult to find proof of its presence. Examinations center on tissue stains, such as Papanicolaou (Pap) staining [78] and Lugol’s staining. In particular, the Lugol’s staining method has been considered effective because the inner surface of the vagina is covered with stratified squamous epithelial cells and contains abundant glycogen [79,80]. However, staining may also be seen in cells of the oral cavity [81,82], and it is now considered that careful evaluation is required. As serological methods, immunodiffusion and immunoelectrophoresis were previously performed using anti-human vaginal fluid precipitin [34]. However, the serological method is rarely performed given the high-titer antibody is not available.

In contrast, because many bacteria, including lactobacilli, parasitize the vagina, a test method has been developed that targets the 16S-23S rRNA genes of bacteria [83,84]. We previously demonstrated the usefulness of PCR amplification of the 16S rRNA of *Lactobacillus crispatus*, *L. jensenii*, and *Atopobium vaginae* [85]. In addition, there are reports on the identification of vaginal secretions by indigenous bacterial flora using next-generation sequencing [86,87], and on the effects of sexually transmitted diseases and their treatments on the bacterial flora [88,89].

As for saliva and semen, many studies have been conducted using real-time RT-PCR [38,90,91]. We have also performed similar studies [35,41] and have revealed the usefulness of multiplex detection specifically targeting *ESR1*, *SERPINB13*, *KLK13*, *CYP2B7P1*, and *MUC4* genes [92,93] for vaginal fluid identification. Methods using microRNA [45,46,48,49] and DNA methylation [52,53,54,55] are also being examined. Furthermore, we have tried to develop novel methods for vaginal fluid identification by GC-MS targeting of 17β-estradiol (E2-17β) and by LC-MS targeting of small proline-rich protein 3 (SPRR3) and fatty acid-binding protein 5 (FABP5) [94,95].

## 6. Discussion

To identify body fluids, the classic techniques, which include enzymatic, serological, and histological methods using a microscope, have been the primary tools for many years. In particular, immunodiffusion and immunoelectrophoresis using high-specificity antibody were effectively used for body fluid identification when the samples were sufficiently secured and relatively fresh. However, with the recent remarkable progress in DNA typing methods, the number of test samples is increasing, as well as the need to analyze very small amounts of samples or old, poorly preserved samples. Therefore, body fluid identification with quicker and more accurate methods are more essential than ever before. Immunochromatographic methods are inevitably being introduced. In addition, molecular biological methods are being developed to keep pace with the DNA detection level. As a result, various new markers, t distinct from conventional markers, have been defined for each body fluid, and studies have been conducted to determine their feasibility for practical application.

Table 1 outlines the methods used to identify the various body fluids. For the body fluids often examined in relation to sexual crimes, the most active research is being conducted into the development of methods using gene-targeted mRNA, microRNA, or DNA methylation. Many researchers have been working on the practical application of mRNA methods for several decades, but the problem of mRNA stability remains [96,97]. Currently, microRNAs must be evaluated by the expression patterns of several markers [48,49,50], whereas for DNA methylation, a relatively large amount of template DNA is required [98]. Therefore, it may be difficult to perform an independent evaluation from a small sample or from stains containing multiple body fluids. To solve these problems, a method for simultaneously extracting RNA and DNA from the same sample and for identifying body fluid by mRNA, microRNA, and DNA methylation, along with STR typing, is also being developed [42,99,100,101,102,103]. Semen and vaginal fluid have often been the targets of sexual crime investigations, but in recent years, in addition to saliva, the need has grown to identify the skin fragments of suspects that become detached during the assault, which are called touch samples. The number of research reports regarding this sample type is increasing [104,105,106,107,108,109,110]. Furthermore, methods using Raman spectroscopy [111] or Fourier-transform infrared spectroscopy [112] have been reported as non-destructive methods for analysis that do not even consume a minute amount of residual sample, although further studies are required before they can be practically applied.

In this review of the current methods for identifying body fluids related to sexual crime, we have discussed various methods currently in use, as well as the latest reports regarding the development of new methods, focusing on saliva, semen, and vaginal secretions. In criminal investigations, body fluid identification is an essential task in correctly applying the obtained results of DNA typing, and thus further research and development are essential.

## Figures and Tables

**Table 1 diagnostics-10-00693-t001:** Assays for the identification of body fluids.

Body Fluid	Enzymatic	Serological	Microscopic	Molecular Biological	Micro-Biological	Other
Saliva	Colorimetry Phadebas^®^ SALIgAE^®^ Blue starch agarose plate method [30,31,32,33]	Immunodiffusion Immunoelectrophoresis ELISA Immunochromatography RSIDTM-Saliva [28,34,35,36]		mRNA microRNA DNA methylation [37,38,39,40,41,42,43,44,45,46,47,48,49,50,51,52,53,54,55,56]	Oral bacteria [57,58,59,60]	
Semen	Acid phosphatase test [61,62]	Immunodiffusion Immunoelectrophoresis Immunochromatography SPERM HY-LITERTM SERATEC®PSA Semiquant RSIDTM-Semen [34,67,68,69,70,71,72,73,74,75]	Baecchi staining Corin-Stockis staining Oppitz staining Hematoxylin & eosin staining [62,63,64,65,66]	mRNA microRNA DNA methylation [37,38,39,40,45,46,48,49,50,51,52,53,54,55,56,76,77]		
Vaginal fluid		Immunodiffusion Immunoelectrophoresis [34]	Papanicolaou staining Lugol’s staining [79,80,81,82]	mRNA microRNA DNA methylation [35,38,41,45,46,48,49,52,53,54,55,90,91,92,93]	Vaginal bacteria [83,84,85,86,87,88,89]	GC-MS LC-MS [94,95]

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
