# Peer review of "Current Methods for Body Fluid Identification Related to Sexual Crime: Focusing on Saliva, Semen, and Vaginal Fluid"

_diagnostics, 2020, doi:10.3390/diagnostics10090693_

Round 1

Reviewer 1 Report

This review aims to analyze the current methods for identifying body fluids related to sexual crime. Particularly, the authors try to discuss both conventional methods and the latest serological and molecular biological methods, focusing on saliva, semen, and vaginal fluid.
Even if this review could be useful for the forensic science community, I believe that several modifications should be made to the text before the publication. I have reported all suggestion in the specific box.

The major concern about this review is the methodology. In general, the reviews must be carried out in an objective, rigorous and meticulous methodology and it must include a search strategy that allows knowing the quality of the articles, the reproducible and explicit selection criteria, and the synthesis and interpretation of the results. In this review, I do not know the methodology that the authors have followed to select certain papers.

The “Abstract” section should be improved. First of all, the review’s aims are not presented. Moreover, the research strategy should be clarified. Please, check it.

The "Introduction" section should be revised. in the two initial paragraphs (lines 28-39 and lines 40-47) the authors discuss the DNA analysis that it is not completely functional to the review's aims. Moreover, for several information no references were inserted. I suggest shortening these paragraphs, summarizing all in one paragraph. Furthermore, the authors missed introducing the importance of the determination of the "nature of the trace" in the forensic field. At this regard, the authors should introduce one of the most important problems related to "touch DNA". They inserted a short consideration in the discussion section, while I believe that it is important to insert it in the introduction. Recently, at this regard, several papers have been published, highlighting the possibility to release a sufficient amount of cells with a simple and very short touch on a garment (you can read these references: - Sessa et al. "Touch DNA: Impact of handling time on touch deposit and evaluation of different recovery techniques: An experimental study. DOI: 10.1038/s41598-019-46051-9;
- Ruan et al. "Investigation of DNA transfer onto clothing during regular daily activities". DOI: 10.1007/s00414-017-1736-x
- Neckovic et al. "Investigation of direct and indirect transfer of microbiomes between individuals" DOI: 10.1016/j.fsigen.2019.102212).
I believe that it could be very useful for the readers to insert a new paragraph about this argument: it represents a thematic that should be focused as an important concern of the forensic field. Please, insert it.
Moreover, at the end of the introduction, it should be important to improve the review's aims.

Before each section, I suggest inserting a short section (Methods), describing the database and the main terms used for the literature review. Moreover, it should be important to indicate the start date and the end date of the research.

Each section should be improved. Particularly, the authors should analyze each new application (miRNAs, mass spectrometry, etc..) for each biological fluid. Even if the new methods could be not validated, I believe that it is important to describe the attempt to perfect new methods.

Finally, I suggest improving the last part of the discussion section inserting several consideration on the future trends of this forensic field.

Author Response

Response to Reviewer #1

We are grateful for your valuable comments and suggestions regarding our manuscript. Our responses to your comments are shown below.

Major comments:

The major concern about this review is the methodology. In general, the reviews must ~~~ select certain papers.

Response:

Thank you for the comment. Please let me explain about this review. This is not a journal for forensic specialists, so we thought to introduce the current examination of objects related to biological samples and also to make it easier to understand, including importance of clarifying the origin of extracted DNA. Therefore, the cited papers may have been limited to our experience and related papers. We apologize for the structure of manuscript and lack of other important articles. We would appreciate your understanding.

Comments:

The “Abstract” section should be improved. First of all, the review’s aims are not presented. Moreover, the research strategy should be clarified. Please, check it.

Response:

In accordance with your comments, we have added the following sentence:

“This review introduces the importance of developing precise methods for body fluids identification.” (Line 14-15)

Comments:

The "Introduction" section should be revised. ~ summarizing all in one paragraph.

Response:

As described in our response to Major comments, many readers of this journal may have little forensic knowledge. So, we would like to leave the sentences if possible.

Comments:

Furthermore, the authors missed introducing the importance of the determination of the "nature of the trace" ~ it should be important to improve the review's aims.

Response:

Thank you for the nice suggestion and suggesting relevant references. However, as the title shows, we would like to focus on saliva, semen, and vaginal fluid in this paper. We would appreciate your understanding.

We have added the following three papers to the references in Discussion section.

“Sessa, F.; Salerno, M.; Bertozzi, G.; Messina, G.; Ricci, P.; Ledda, C.; Rapisarda, V.; Cantatore, S.; Turillazzi, E.; Pomara, C. Touch DNA: impact of handling time on touch deposit and evaluation of different recovery techniques: An experimental study. Sci. Rep. 2019, 9, 9542.”

“Ruan, T.; Barash, M.; Gunn, P.; Bruce, D. Investigation of DNA transfer onto clothing during regular daily activities. Int. J. Legal Med. 2018, 132, 1035-1042.”

“Neckovic, A.; 1, van Oorschot, R.A.H.; Szkuta, B.; Durdle, A. Investigation of direct and indirect transfer of microbiomes between individuals. Forensic Sci. Int. Genet. 2020,45, 10212.”

Comments:

Before each section, ~~~ and the end date of the research.

Response:

Thank you for the suggestion. However, each section itself is very short, so I would like to keep this style if possible.

We wish to express our appreciation to the Reviewer for their insightful comments, which have helped us considerably in revising the paper.

Reviewer 2 Report

Line 192-194: These sentences seem to refer to vaginal fluid. However, they start with 'As for saliva and semen.' Do ESR1 and other genes in line 194 refer to the identification of vaginal fluid? Please clarify the sentences.

Regarding Table 1: In the vaginal fluid section, there is 'Immunodiffusion Immunoelectrophores  is.'  The 'is' should be connected with 'Immunoelectorophres' as one word.

Author Response

Response to Reviewer #2

We are grateful for your helpful comments regarding our manuscript. Our responses to your comments are shown below.

Comments:

Line 192-194: These sentences seem to refer to vaginal fluid. However, they start with 'As for saliva and semen.' Do ESR1 and other genes in line 194 refer to the identification of vaginal fluid? Please clarify the sentences.

Response:

We apologize for unclear sentence. Yes, the genes in line 194 refer to the identification of vaginal fluid. In accordance with your comment, we have added the following sentence after the [91, 91],

“for vaginal fluid identification” (Line 195-196)

Comments:

Regarding Table 1: In the vaginal fluid section, there is 'Immunodiffusion Immunoelectrophores  is.'  The 'is' should be connected with 'Immunoelectorophres' as one word.

Response:

We apologize for these editing errors. We have corrected them in Table 1.

Reviewer 3 Report

The manuscript entitled “Current methods for body fluid identification related to sexual crime: focusing on saliva, semen, and vaginal fluid” is a concise review of the main methods that forensic laboratories are currently using to identify the different body fluids that could be found on a sexual crime, i.e. saliva, semen and vaginal fluid. It comprises the most common and known assays available for each body fluid, organized according to the analysis method they are based on. The manuscript also summarizes the assays of the latest methods that are now under exploration, such as the analysis of mRNA, microRNA and DNA methylation.

This is a valuable work for the forensic community, that could be a good starting point for researches who want to begin work on body fluid identification.

I only suggest a few minor comments:

- Lines 32-33: include the acronym of "Combined DNA Index System" in brackets, "Combined DNA Index System (CODIS) DNA database".

- Lines 35-36: when describing the autosomal loci included on GlobalFiler, it seems that "the autosomal 21 STR locus" refers to the loci included in the expanded CODIS core loci. However, GlobalFiler includes the 20 autosomal STR loci of extended CODIS and the extra loci SE33. If the sentence does not mean this, the article "the" should be removed. On the other hand, the plural for “locus” is “loci”. Thus, a possible alternative for this could be "… amplifies 21 autosomal STR loci". 

- Lines 36-37: in the second part of this sentence I have two comments. Firstly, instead of "discriminating power" it is preferable to say "power of discrimination". On the other hand, when it says "frequency of occurrence" I guess it refers to the probability of identity, but it is not clear. Could you rewrite this second part of the sentence (lines 36 to 38) to make it easier to understand?

- Lines 37-38: instead of using quadrillion and trillion, or in addition to them, it would be advisable to write them numerically to avoid possible confusions (for instance, 1024 or whatever it is).

- Paragraph 87-94: I would suggest to include some references related to ABO blood grouping and its analysis in other tissues.

- Lines 124-125: to make it clearer I would specify that what RSID detects is also alpha-amylase antigen.

Author Response

Response to Reviewer #3

We are grateful for your helpful comments regarding our manuscript. Our responses to your comments are shown below.

Comments:

- Lines 32-33: include the acronym of "Combined DNA Index System" in brackets, "Combined DNA Index System (CODIS) DNA database".

Response:

In accordance with your comment, we have added the (CODIS) as follows,

Combined DNA Index System (CODIS) DNA database"

Comments:

- Lines 35-36: when describing the autosomal loci included on GlobalFiler, it seems that "the autosomal 21 STR locus" refers to the loci included in the expanded CODIS core loci. However, GlobalFiler includes the 20 autosomal STR loci of extended CODIS and the extra loci SE33. If the sentence does not mean this, the article "the" should be removed. On the other hand, the plural for “locus” is “loci”. Thus, a possible alternative for this could be "… amplifies 21 autosomal STR loci".

Response:

Thank you for suggesting the appropriate expression. We have revised the sentence as follows,

“the GlobalFilerTM reagent amplifies 21 autosomal 21 STR loci,” (Line 36-37)

Comments:

- Lines 36-37: in the second part of this sentence I have two comments. Firstly, instead of "discriminating power" it is preferable to say "power of discrimination". On the other hand, when it says "frequency of occurrence" I guess it refers to the probability of identity, but it is not clear. Could you rewrite this second part of the sentence (lines 36 to 38) to make it easier to understand?

Response:

We apologize for the inappropriate sentence. In accordance with your comment, we have revised the sentence (lines 36-39) as follows,

“In particular, the GlobalFilerTM reagent amplifies 21 autosomal STR loci, DYS391 locus, Y indel locus, and amelogenin locus. Fujii et al. [3] reported that the probability of identity of these 21 autosomal loci obtained using GlobalFiler was 1.84 × 10−25 in Japanese population, which was remarkably higher than that (1.8 × 10−17) of the 15 autosomal loci obtained using AmpFLSTR Identifier kit (Life Technologies) [3].” (Line 36-40)

Comments:

- Lines 37-38: instead of using quadrillion and trillion, or in addition to them, it would be advisable to write them numerically to avoid possible confusions (for instance, 1024 or whatever it is).

Response:

In accordance with your comment, we have revised them, as mentioned above.

Comment:

Paragraph 87-94: I would suggest to include some references related to ABO blood grouping and its analysis in other tissues.

Response:

In accordance with your comment, we have added some papers as follows,

“Coombs, R.R.A.; Dodd, B. Possible application of the principle of mixed agglutination in the identification of blood stains. Med. Sci. Law. 1961, 1, 359-377.”

“Pereira, M. ABO and Lewis typing of semen, saliva and other body fluids. Haematologia, 1984,17, 317-322.”

“Miyasaka, S.; Yoshino, M.; Sato, H.; Miyake, B.; Seta, S. The ABO blood grouping of a minute hair sample by the immunohistochemical technique. Forensic Sci. Int. 1987, 31, 85-98.”

Comment:

Lines 124-125: to make it clearer I would specify that what RSID detects is also alpha-amylase antigen.

Response:

In accordance with your comment, we have revised the sentence as follows,

“Another saliva confirmatory test is the RSIDTM-Saliva (Independent Forensics, Lombard, IL) kit for forensic science, which involves immunochromatography for human salivary α-amylase.”

Round 2

Reviewer 1 Report

The authors have addressed all of my comments.